# A Novel Method for Visualizing Melanosome and Melanin Distribution in Human Skin Tissues

**DOI:** 10.3390/ijms21228514

**Published:** 2020-11-12

**Authors:** Chikako Yoshikawa-Murakami, Yuki Mizutani, Akemi Ryu, Eiji Naru, Takashi Teramura, Yuta Homma, Mitsunori Fukuda

**Affiliations:** 1KOSÉ R&D France, KOSÉ Corporation, 5 Avenue Lionel Terray, 69330 Meyzieu, France; c-yoshikawa@kose.co.jp (C.Y.-M.); t-teramura@kose.co.jp (T.T.); 2Research Laboratories, KOSÉ Corporation, 48-18 Sakae-cho, Kita-ku, Tokyo 114-0005, Japan; a-ryu@kose.co.jp (A.R.); e-naru@kose.co.jp (E.N.); 3Laboratory of Membrane Trafficking Mechanisms, Department of Integrative Life Sciences, Graduate School of Life Sciences, Tohoku University, Aobayama, Aoba-ku, Sendai, Miyagi 980-8578, Japan; y-homma@biology.tohoku.ac.jp

**Keywords:** human skin, keratinocyte, melanin distribution, melanocyte, melanosome, M-INK, skin phototype, three-dimensional observation

## Abstract

Melanin incorporated into keratinocytes plays an important role in photoprotection; however, abnormal melanin accumulation causes hyperpigmentary disorders. To understand the mechanism behind the accumulation of excess melanin in the skin, it is essential to clarify the spatial distribution of melanosomes or melanin in the epidermis. Although several markers have been used to detect melanosomes or melanin, no suitable markers to determine the precise localization of melanin in the epidermis have been reported. In this study, we showed that melanocore-interacting Kif1c-tail (M-INK), a recently developed fluorescent probe for visualizing mature melanosomes, binds to purified melanin in vitro, and applied it for detecting melanin in human skin tissues. Frozen skin sections from different phototypes were co-stained for the hemagglutinin (HA)-tagged M-INK probe and markers of melanocytes or keratinocytes, and a wide distribution of melanin was observed in the epidermis. Analysis of the different skin phototypes indicated that the fluorescent signals of HA-M-INK correlated well with skin color. The reconstruction of three-dimensional images of epidermal sheets enabled us to observe the spatial distribution of melanin in the epidermis. Thus, the HA-M-INK probe is an ideal tool to individually visualize melanin (or melanosome) distribution in melanocytes and in keratinocytes in skin tissues.

## 1. Introduction

Melanin is one of the main factors responsible for human skin color. It is synthesized by melanogenic enzymes such as tyrosinase within melanosomes, which are specialized organelles in melanocytes, and is finally transported to and distributed in neighboring keratinocytes. Melanosomes transferred to the keratinocytes sequester the nucleus to protect against ultraviolet light, which induces nuclear DNA damage [1,2,3,4]. Despite its important role in photoprotection, however, abnormal accumulation of melanin causes hyperpigmentary disorders such as melasma, senile lentigo, and freckles [2]. To prevent the accumulation of excess melanin in the skin, it is essential to clarify the spatial distribution of melanosomes in the epidermis, which helps us to understand the precise process of melanosome transfer from melanocytes to keratinocytes and of melanosome degradation in keratinocytes.

To detect melanosomes or melanin, several methods targeting a melanosome-specific protein or the reducing ability of melanin have been used. One example of this is the use of Fontana–Masson silver staining, which produces black silver spots that are reaction products of the reducing group of melanin [5]. This approach has been widely used for detecting melanosomes or melanin histochemically [6,7,8]. However, this method detects not only melanin but also lipofuscin and other reducing groups [9]. Furthermore, it is impossible to use this approach to observe stained spots together with several proteins simultaneously, and to analyze their three-dimensional (3D) distribution by confocal microscopy. Another method widely used for detecting melanosomes or melanin histochemically is immunostaining for Pmel (also known as Pmel17 and gp100), a transmembrane glycoprotein present in melanosomes [10]. However, the antibody against Pmel is unable to detect mature black melanosomes because Pmel is surrounded by a large amount of melanin, which prevents antibody access in mature melanosomes [11]. Moreover, antibodies against melanogenic enzymes such as tyrosinase and tyrosinase-related protein 1 (Tyrp1) cannot be applied to stain transferred melanosomes in keratinocytes because they are readily degraded in keratinocytes [12]. Therefore, Pmel and melanogenic enzymes are inappropriate markers to monitor the series of processes of melanosome transfer, dispersion, and degradation in keratinocytes.

To resolve these issues, we recently developed a novel tool called melanocore-interacting Kif1c-tail (M-INK), which can recognize mature melanosomes in cultured cells in a melanin-content-dependent manner via an unknown mechanism [13]. In this study, we present evidence that M-INK directly binds to purified melanin. We also applied the newly developed M-INK probe to human skin tissue and, for the first time, established an easy-to-use protocol to visualize melanin (or mature melanosomes) even in skin tissue. Finally, we report the precise distribution of melanin in keratinocytes and melanocytes using reconstructed 3D images.

## 2. Results

### 2.1. M-INK Recognizes Both Mature Melanosomes and Purified Melanin

Previously, we reported that melanin-containing melanosomes in cultured melanocytes were specifically detected with purified T7-glutathione *S*-transferase (GST)-tagged M-INK [13]. However, the original M-INK probe (T7-GST-M-INK) needs to be purified from COS-7 cells and only a small amount of protein was obtained from several 10 cm dishes. To improve the protocol for visualizing melanosomes with M-INK, we constructed hemagglutinin (HA)-tagged M-INK (HA-M-INK), expressed it in COS-7 cells, and attempted to use their cell lysates containing HA-M-INK as a melanin or mature melanosome marker without purification. First, we investigated the ability of HA-M-INK to recognize melanin-containing melanosomes using melanin-producing B16 melanoma cells and non-melanin-producing NB1RGB fibroblast cells (Figure 1A). These cells were incubated with HA-M-INK-containing cell lysates (hereafter referred to as HA-M-INK lysates), and HA-M-INK signals were visualized with anti-HA tag antibody. The results in B16 melanoma cells show marked colocalization between HA-M-INK signals identified by immunofluorescence staining and melanin-containing melanosomes on bright-field images (Figure 1A, red insets in the top row), consistent with our previous results obtained using purified T7-GST-M-INK [13]. On the other hand, no fluorescent signals were detected in NB1RGB fibroblast cells (Figure 1A, bottom row). Moreover, untransfected COS-7 cell lysates (a negative control) showed no obvious signals under our experimental conditions (data not shown).

We also observed fluorescent and bright-field images of B16 melanoma cells with a focus on melanin and Pmel, a typical melanosomal membrane-bound glycoprotein used as an immature melanosome marker. In contrast to HA-M-INK described above, fluorescent signals from the Pmel antibody were not well colocalized with melanin-containing melanosomes on bright-field images (Figure 1B, red insets in the top row). Specifically, from bright-field microscopic observations, the Pmel antibody often recognized dotted structures in the areas devoid of melanin, indicating that Pmel is not a good marker for mature melanosomes (Figure 1B, white insets in the top row). We also confirmed that Pmel-positive signals were not colocalized with HA-M-INK-positive signals, which showed black mature melanosomes in B16 melanoma cells (Appendix A).

Since M-INK can recognize melanin-containing mature melanosomes, but not immature melanosomes (Figure 1A) [13], we hypothesized that M-INK directly binds to melanin itself. To test this hypothesis, we investigated the in vitro melanin-binding activity of purified T7-GST-M-INK and T7-GST alone as a negative control by using purified melanin. After incubating T7-GST-M-INK (or T7-GST) with melanin for 20 min, the insoluble melanin was removed by centrifugation, and the levels of proteins remaining in the supernatant were determined (Figure 2A,B). As expected, T7-GST-M-INK in the supernatant was drastically decreased compared with that in control T7-GST (*p* < 0.001). By contrast, the amount of T7-GST protein in the supernatant was largely unchanged between before and after incubation with the purified melanin, indicating that the T7-GST tag itself has no or little melanin-binding activity. These results taken together indicate that the M-INK probe recognizes mature melanosomes in cultured melanocytes, most likely by binding to melanin within melanosomes.

### 2.2. HA-M-INK Recognizes Melanin or Black Mature Melanosomes in Human Skin Tissue

Next, we investigated whether HA-M-INK recognizes melanin (or black mature melanosomes) even in human skin samples, in addition to cultured cells. Skin tissue sections were treated with HA-M-INK lysates or control COS-7 cell lysates, followed by immunohistochemical (IHC) staining with the anti-HA tag antibody; then, the stained samples were observed by confocal microscopy. HA-M-INK staining with high fluorescence intensity on IHC images colocalized well with black basal melanin on bright-field images (Figure 3, white arrowheads in the yellow insets in the top row). Moreover, HA-M-INK staining overlapped well with the supranuclear melanin cap in the spinous layer (Figure 3, white arrow in the red insets in the top row). On the other hand, treatment with control COS-7 cell lysates did not result in any positive staining signals from the black spots of melanin, which were clearly observed in bright-field images (Figure 3, arrow and arrowheads in the bottom row).

To determine whether the HA-M-INK signals in the skin samples described above correspond to melanin located within melanocytes or to those transferred from melanocytes to keratinocytes, skin sections were co-treated with HA-M-INK lysates and antibodies against Pmel, c-KIT protein (a melanocyte surface receptor), and desmoplakin 1 (DP-1, an intercellular adhesion molecule in keratinocytes). The results show that HA-M-INK colocalized with c-KIT-positive melanocytes as well as DP-1-positive keratinocytes in the vicinity of melanocytes (Figure 4A, red inset in the top row). By contrast, most of the Pmel-positive signals were observed in the c-KIT-positive melanocytes; however, only a few positive signals were detected in the DP-1-positive keratinocytes (Figure 4B, red inset in the bottom row), indicating that the anti-Pmel antibody fails to efficiently detect melanin transferred to keratinocytes. Thus, our results indicate that the newly developed HA-M-INK can individually detect melanin present in melanocytes and keratinocytes.

### 2.3. Correlation between Skin Phototypes and Fluorescence Signals by HA-M-INK

Since human skin color phenotypes are known to be determined by the melanin content and distribution in keratinocytes [14,15], we compared the IHC distribution of HA-M-INK between skin samples of different Fitzpatrick phototypes. Three healthy volunteers were recruited, one each with Fitzpatrick phototype II, III, or V, and skin samples were surgically obtained from them. The L* value of their skin surface was then measured and their skin sections were treated with HA-M-INK lysates for microscopic observations (Figure 5). The skin sample of phototype II was mostly negative for HA-M-INK, consistent with the bright-field observation that black melanin spots were not visible (Figure 5, top row). The sample of phototype III had regions positive for HA-M-INK staining, with strong signals colocalizing with melanin caps in the basal and spinous layers (Figure 5, middle row). The sample of phototype V showed the strongest signal intensities, with discrete high-intensity regions overlapping with melanin caps that were widely distributed across the basal and spinous layers (Figure 5, bottom row). The fluorescence intensity of HA-M-INK measured over different epidermal layers, excluding the stratum corneum, seemed to increase with decreasing skin L* values, that is, with higher Fitzpatrick phototypes (Table 1).

### 2.4. Three-Dimensional (3D) Reconstruction of Epidermal Tissue Structure

Epidermal sheets prepared from a skin sample corresponding to Fitzpatrick phototype V were subjected to IHC staining using HA-M-INK lysate and markers for melanocytes (c-KIT) and keratinocytes (DP-1). Confocal microscopy images were taken from the dermal side and obtained from the epidermal basement membrane to the spinous layer (Figure 6). The dark regions in the images negative for 4’,6-diamidino-2-phenylindole (DAPI) represent epidermal structures protruding into the papillary dermis (Figure 6, asterisks). When we observed the lower surface of the basal layer, HA-M-INK-positive signals were localized in the DP-1-positive basal layer keratinocytes (Figure 6A). We then subjected the epidermal sheet labeled with HA-M-INK and c-KIT to serial section confocal imaging (0.5-μm z-step), and the fluorescent images collected from the basal to the spinous layers were superimposed. The resulting images show c-KIT-positive melanocytes with branching dendrites as well as the presence of HA-M-INK-positive areas within c-KIT-positive melanocytes (Figure 6B, red inset). Moreover, HA-M-INK-positive regions were present in keratinocytes in contact with neighboring melanocytes as well as in isolated keratinocytes located far from melanocytes (Figure 6B, yellow inset).

Finally, using the fluorescent images obtained above, melanin, melanocyte, and keratinocyte structures were reconstructed three-dimensionally (Figure 6C). The 3D reconstructed images showed that c-KIT-positive melanocytes extended their dendrites upward towards the epidermal spinous layer. Moreover, the basal layer showed a nonuniform spatial distribution of HA-M-INK-positive signals. Thus, the HA-M-INK probe would provide a powerful means of performing simultaneous 3D structural analysis of melanocytes, keratinocytes, and melanin.

## 3. Discussion

We previously reported that T7-GST-tagged M-INK recognizes both melanosomes and melanocores in cultured melanocytes via an unknown mechanism [13]. In the present study, we provided the first evidence that T7-GST-M-INK directly binds to purified melanin (Figure 2). We then developed HA-M-INK as a new probe and established a protocol to use it to efficiently and easily visualize melanin (or black matured melanosomes) in cultured cells and human skin tissues. The main advantage of this protocol is that the new HA-M-INK probe in crude lysates of transfected COS-7 cells can be directly applied to immunofluorescence analysis without further purification, in contrast to the original T7-GST-M-INK probe, which needs to be affinity-purified before use. HA-M-INK successfully visualized melanin-containing melanosomes in melanoma cells (Figure 1), melanin in the basal keratinocytes, and melanin cap in spinous keratinocytes (Figure 3). To remove non-specific signals of COS-7 cell lysates, we always used untransfected COS-7 cell lysates as a negative control, which can confirm the specific signals of HA-M-INK (Figure 3, bottom panels, and data not shown).

Two well-known techniques for detecting melanosomes and melanin in cultured cells and skin sections are Fontana–Masson staining and immunostaining using antibodies against melanosomal proteins such as tyrosinase, Tyrp1, and Pmel. Compared with these techniques, the HA-M-INK probe developed here has several advantages. First, it can visualize melanin “three-dimensionally,” in contrast to Fontana–Masson staining [5]. Second, it can visualize melanin even “in keratinocytes”. In fact, anti-tyrosinase and anti-Tyrp1 antibodies failed to recognize transferred melanosomes in keratinocytes [12,13,16], and anti-Pmel antibody cannot recognize black mature melanosomes ([13] and this study). Third, it allows for “simultaneous visualization of melanin and other cell components” when combined with their antibodies even in human skin tissues (Figure 4 and Figure 6). Specifically, HA-M-INK can visualize melanin present in c-KIT-positive melanocytes, as well as those transferred to DP-1-positive keratinocytes (Figure 4A and Figure 6A).

The combined use of antibodies against Pmel and Tyrp1 provides an effective method for identifying different stages of melanosome maturation in melanocytes because Pmel is a structural component already formed in stage I and II melanosomes, whereas melanogenic enzymes such as tyrosinase and Tyrp1 are enriched in stage III and IV melanosomes [17]. Adding this newly developed HA-M-INK probe, which binds to melanin itself and detects matured melanosomes in both melanocytes and keratinocytes, deepens our understanding of melanosome maturation in melanocytes and the process of melanosome transfer to keratinocytes. Moreover, the combination of HA-M-INK and anti-Pmel antibody may help to determine the distributions and amounts of both undegraded and degraded (or partially degraded) melanosomes in keratinocytes (Figure 4 and Appendix A).

Taking advantage of the unique biochemical features of HA-M-INK described above, it was successfully applied to confocal microscopy analysis of an epidermal sheet to observe two- dimensional (2D) and 3D distributions of epidermal melanin. The resulting images show that melanocytes were distributed along the basal layer and extended their dendrites toward neighboring keratinocytes. These results are consistent with those of a previous study involving 3,4-dihydroxyphenylalanine (L-Dopa) staining of normal human skin samples, which showed that L-Dopa-positive melanocytes were located between basal cells, and their dendrites extended into the intercellular space between keratinocytes [18]. Moreover, the images in the present study show that melanin was present in keratinocytes both close to and far from c-KIT-positive melanocytes (Figure 6B). Considering that one melanocyte contacts 30 to 40 neighboring keratinocytes [19], the transfer of melanin to remote keratinocytes is likely to occur through dendrite extension of melanocytes.

Given the potential of HA-M-INK to detect melanin in keratinocytes, we were able to directly compare the melanin distribution and content between skin samples collected from volunteers with phototypes II, III, and V: skin with higher phototypes was associated with higher HA-M-INK fluorescence (Figure 5), although we need to test more samples to support this conclusion. This finding is consistent with previous reports describing that the amount and distribution of epidermal melanin were associated with skin color [14,15]. In this study, our fluorescent intensity of HA-M-INK analysis was confined to the epidermis except stratum corneum, even though its fluorescence values were markedly elevated in the stratum corneum, particularly in phototype V samples. Since several antibodies sometimes recognize stratum corneum non-specifically, based on our experience, further analysis is needed to clarify whether the high fluorescence in the stratum corneum is due to the binding of HA-M-INK to corneal melanin or nonspecific adsorption of HA-M-INK to the stratum corneum. The difference in skin color is thought to depend on the regulation of several molecules involved in melanin synthesis, transfer, and storage in keratinocytes [8,20,21], and the rate of melanosome degradation in epidermal keratinocytes [22,23]. We thus think that HA-M-INK would also serve as an effective tool for elucidating the molecular mechanisms underlying melanin transfer and storage in keratinocytes and their roles in skin color variation.

In conclusion, we have developed a new melanin probe, named HA-M-INK, and established an easy-to-use protocol for visualizing 2D and 3D distributions of melanin (or mature melanosomes) as well as for quantitatively assessing melanin in the epidermis by immunofluorescent staining in human skin tissues. The future application of the HA-M-INK probe to the study of pigmentary disorders such as senile lentigo, melasma, and freckles should provide valuable insights into the precise spatial distribution of melanin and its interaction with other molecules, clarifying the etiology of these conditions and how to prevent and treat them.

## 4. Materials and Methods

### 4.1. Materials

The following antibodies used in this study were obtained commercially: anti-HA rat monoclonal antibody (clone 3F10) (#11867423001; Roche Diagnostics, Mannheim, Germany); Alexa 488-conjugated goat anti-rat IgG(H+L) (#ab150165; Abcam, Cambridge, UK); anti-human c-Kit (K963) rabbit IgG (#18101; IBL, Fujioka, Japan); Alexa Fluor 488-conjugated goat anti-mouse IgG(H+L) (#A11029); Alexa Fluor 568-conjugated goat anti-rabbit IgG(H+L) (#A11011; Thermo Fisher Scientific, Waltham, MA, USA); anti-Pmel mouse monoclonal antibody (anti-human melanosome, clone HMB45) (#M0634; Dako, Carpinteria, CA, USA); anti-desmoplakin 1 guinea pig polyclonal, serum (#DP-1; PROGEN, Wayne, PA, USA); and Alexa Fluor 647 goat anti-guinea pig IgG (H+L) highly cross-absorbed (#A-21450; Thermo Fisher Scientific).

### 4.2. Plasmid Construction

The cDNA fragment encoding M-INK (amino acids 879–1100 of mouse Kif1c) [13] was subcloned into the pEF-HA tag vector (named pEF-HA-M-INK) as described previously [24], and sequences of the insert DNA were verified by sequencing.

### 4.3. Preparation of HA-M-INK

COS-7 cells were cultured in Dulbecco’s modified Eagle’s medium supplemented with 10% fetal bovine serum (FBS), 0.6% glutamine, and 2% NaHCO_3_ at 37 °C under a 5% CO_2_ atmosphere. Plasmids (pEF-HA-M-INK) were transfected into COS-7 cells (subconfluent in one 10 cm dish) by Lipofectamine 2000 reagent (Thermo Fisher Scientific), in accordance with the manufacturer’s instructions. One day after transfection, the cells were disrupted with sterilized distilled water. After removing cell debris by centrifugation, the cell lysates were mixed with a 10× phosphate-buffered saline (PBS) solution. The recovered cell lysates were reconstructed to 1× PBS and used as HA-M-INK lysates. These lysates could be stored at −80 °C for several months before use. COS-7 cell lysates without HA-M-INK expression were used as a negative control.

### 4.4. Cell Cultures

NB1RGB human skin fibroblast cells were cultured in Dulbecco’s modified Eagle’s medium supplemented with 10% FBS, 0.6% glutamine, and 2% NaHCO_3_ at 37 °C under a 5% CO_2_ atmosphere. B16 melanoma cells were cultured in Eagle’s medium supplemented with 10% FBS, 0.3% glutamine, and 2% NaHCO_3_ at 37 °C under a 5% CO_2_ atmosphere. For immunofluorescence analysis, these cells were seeded onto glass-bottomed dishes at 3 × 10^5^ cells/mL for 1 day.

### 4.5. Interaction of M-INK with Purified Melanin In Vitro

One milligram of purified melanin (Cat#: M2649; Sigma-Aldrich, St. Louis, MO, USA) was suspended in 100 mL of PBS. The melanin was spun down by centrifugation at 10,000× *g*, and after removal of the supernatant, it was resuspended in 100 μL of PBS. This washing procedure was repeated four times. Recombinant T7-tagged GST and GST-M-INK were expressed in COS-7 cells and purified as described previously [13]. Approximately 1 μg of T7-GST-M-INK (or T7-GST as a control) was incubated for 20 min with or without 100 μg of melanin in 35 μL of PBS containing 0.1% Triton X-100. After centrifugation at 20,000× *g* for 1 min, the supernatant was recovered and analyzed by 12% SDS-polyacrylamide gel electrophoresis (PAGE), followed by immunoblotting with horseradish peroxidase-conjugated anti-T7 tag antibody (1:5,000 dilution; Cat#: 69048; Merck Millipore, Burlington, MA, USA). Immunoreactivity bands were detected using an ECL substrate (Bio-Rad, Hercules, CA, USA) and a chemiluminescence imager (ChemiDoc Touch; Bio-Rad). The band intensity was quantified using Image Lab software (Bio-Rad).

### 4.6. Tissues

Skin samples were obtained from human donor surgery following the provision of informed consent in accordance with applicable ethical guidelines and regulations in CTIBIOTECH (Meyzieu, France). CTIBIOTECH has approval from Committee for the Protection of Persons (CPP SUD EST IV) which is the French Ethics Regulatory Committee with combined authorization from the French Ministry in charge of research for the preparation and conservation of elements derived from the human body (CODECOH) approved on 28/12/2018 (#AC-2018-3243). Normal human skin samples were obtained surgically, 50-year-old female breast skin and 18-, 27-, or 54-year-old female abdominal skin. All donors were evaluated for skin phototype in accordance with the Fizpatrick–Pathak classification using a questionnaire [25].

### 4.7. Skin Color Measurements

We used a spectrophotometer (CM-700d; Konica Minolta Inc., Tokyo, Japan) to measure skin surface color with the L* a* b* color notation system [26]. The L* parameter represents luminance (L* = 0 for absolute black and L* = 100 for absolute white), so lower L* represents darker pigmentation of the skin surface. We measured the L* value of the skin samples as an index of the skin color related to melanin content. Three measurements were performed for each sample.

### 4.8. Preparation of Epidermal Sheets and Skin Specimens

To prepare the epidermal sheets, the fat tissue layer was removed from the skin samples and the trimmed skin specimens were treated with 1500 PU/mL Dispase II (Sigma-Aldrich) in PBS for 40 min at 37 °C to separate the epidermis from the dermis. To prepare the skin sections, the skin samples were embedded in O.C.T. compound (Sakura Finetek Japan Co., Ltd., Tokyo, Japan), frozen at −80 °C, and sectioned into 10-µm-thick specimens.

### 4.9. Immunohistochemistry

NB1RGB cells or B16 melanoma cells were fixed in 4% paraformaldehyde and permeabilized with 0.1% Triton X-100. The epidermal sheets and skin sections described above were fixed in ice-cold 95% ethanol for 30 min. Each sample was immersed in a blocking solution containing 10% newborn calf serum in PBS for 60 min at room temperature and incubated with HA-M-INK lysates (or control COS-7 cell lysates) in the blocking solution at 4 °C overnight. After washing three times with PBS, they were incubated with anti-HA tag antibody (1:400 dilution), anti-Pmel antibody (HMB45; 1:400 dilution), anti-c-Kit antibody (1:100 dilution), or anti-desmoplakin 1 (DP-1) antibody (1:400 dilution) at room temperature for 60 min. The samples were then washed three times with PBS, incubated with DAPI (Cayman Chemical, Ann Arbor, MI, USA) and anti-rat/mouse/rabbit/guinea pig Alexa Fluor 488/568/647 IgG at room temperature for 60 min, and mounted in Mowiol (Merck Millipore).

### 4.10. Confocal Microscopy

The stained samples were examined for fluorescence with a confocal fluorescence microscope (LSM 700 or 800 or 880; Zeiss, Carl Zeiss, Oberkochen, Germany). Darkly pigmented organelles within melanocytes were identified by bright-field microscopy and designated as mature melanosomes. The images were processed with Image J (version 1.53c; National Institutes of Health, Bethesda, MD, USA). Confocal microscopy images were processed and analyzed with IMARIS (version 8.4; Bitplane, Zurich, Switzerland) for 3D image visualization.

### 4.11. Measurement of Fluorescence Intensity

The fluorescence intensity of HA-M-INK in confocal microscopy images (16 bit) was measured by image analysis using Image J. The mean fluorescent intensity was estimated in the epidermis, excluding the stratum corneum, by determining HA-M-INK-positive signal intensity per 1000 μm^2^. The measurements were performed in three areas per image, and the average value was shown as the fluorescence of HA-M-INK. The units are presented as mean grayscale value. The fluorescence intensity for each skin sample, skin phototype, donor age, sex, source, and L* value are shown in Table 1.

## Figures and Tables

**Figure 1 ijms-21-08514-f001:**
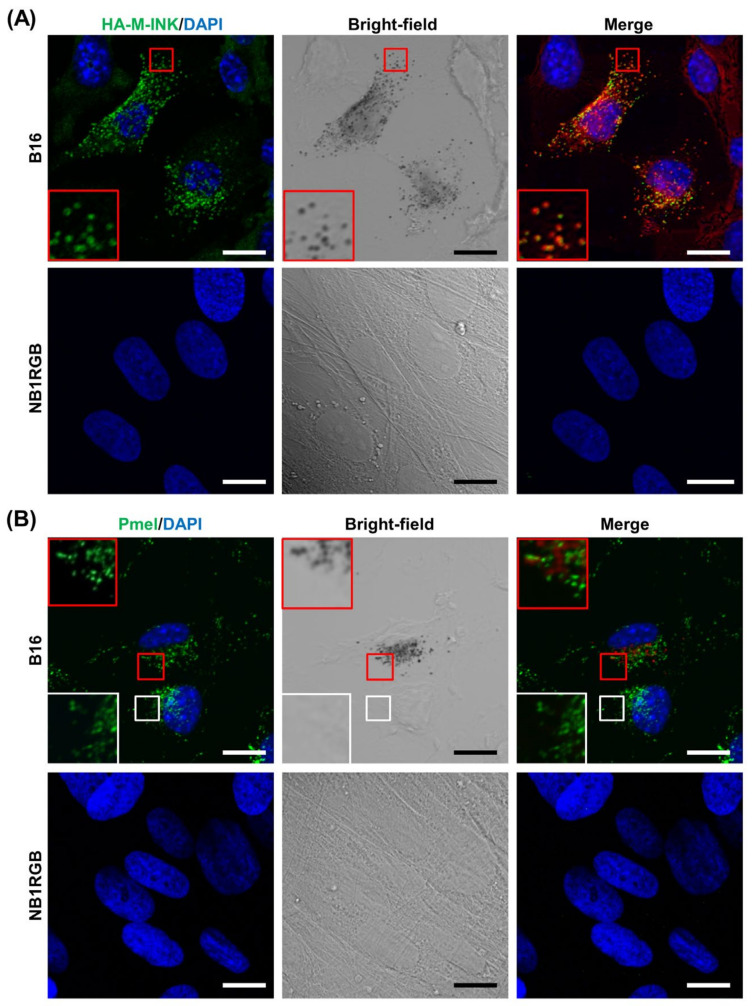
Immunostaining images of B16 melanoma cells and NB1RGB fibroblast cells by hemagglutinin (HA)-tagged melanocore-interacting Kif1c-tail (M-INK) staining. Melanin-containing B16 melanoma cells (top) or non-melanin-producing NB1RGB fibroblasts (bottom) were stained for (**A**) HA-M-INK (green) or (**B**) Pmel (green). 4’,6-diamidino-2-phenylindole (DAPI) was used as a nuclear counterstain (blue). The insets are magnified views of the boxed areas (red and white). The melanosomes in the merged images are presented as pseudo-red-colored signals (red and white insets in the right column). Note that HA-M-INK staining was well colocalized with black mature melanosomes, whereas Pmel was not colocalized with melanin-containing melanosomes. Scale bars = 20 µm.

**Figure 2 ijms-21-08514-f002:**
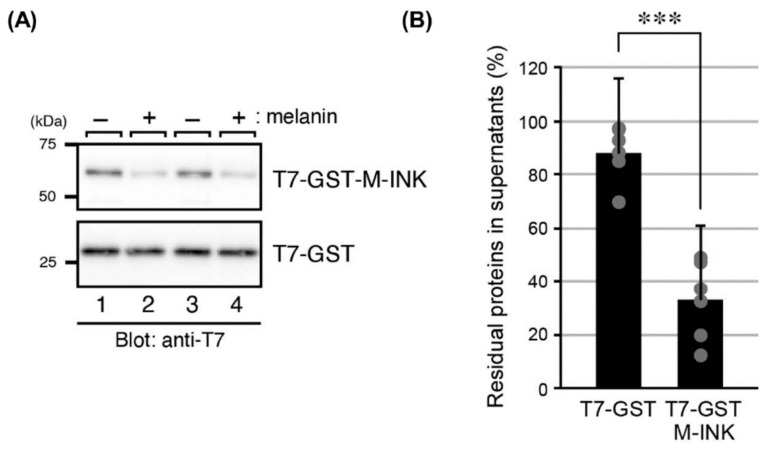
Incubation of T7-glutathione *S*-transferase (GST)-M-INK with purified melanin in vitro. (**A**) Interaction of T7-GST-M-INK with purified melanin in vitro. T7-GST-M-INK or T7-GST (as a negative control) was incubated for 20 min with (+) or without (−) melanin, as described in the Materials and Methods. After removal of insoluble melanin by centrifugation, the supernatant was analyzed by 12% SDS-PAGE, followed by immunoblotting with anti-T7 tag antibody; (**B**) the intensity of the bands shown in (**A**) was quantified. Note that T7-GST-M-INK was efficiently depleted from the supernatant by purified melanin. ***, *p* < 0.001 (Student’s unpaired *t*-test).

**Figure 3 ijms-21-08514-f003:**
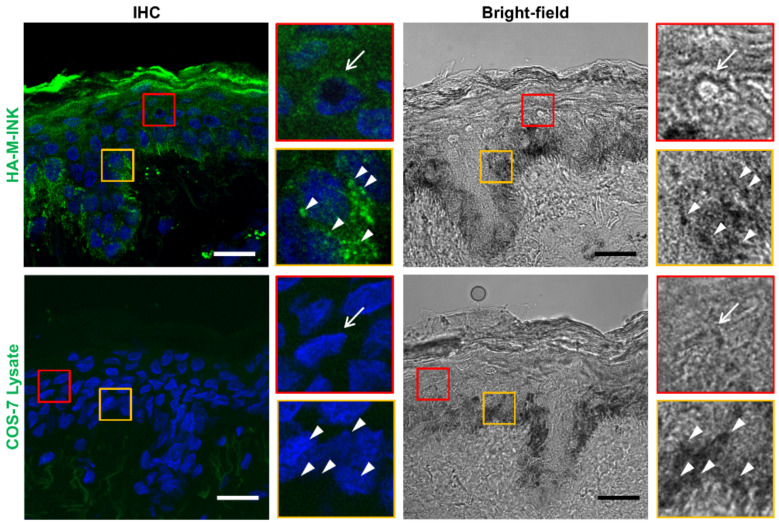
Immunohistochemical (IHC) staining images of melanin in human skin tissue by HA-M-INK staining. Skin sections of phototype IV were stained with HA-M-INK lysates or COS-7 lysates as a control (green), and with DAPI for nuclear counterstaining (blue). The red insets are magnified views of the boxed areas to show the melanin distribution (melanin cap) in the spinous layer (white arrows). The yellow insets are magnified views of the boxed areas to show the melanin distribution in the basal layer (white arrowheads). HA-M-INK staining colocalized with black spots of melanin on bright-field images. Note that untransfected COS-7 cell lysates showed no positive signals. Scale bars = 20 µm.

**Figure 4 ijms-21-08514-f004:**
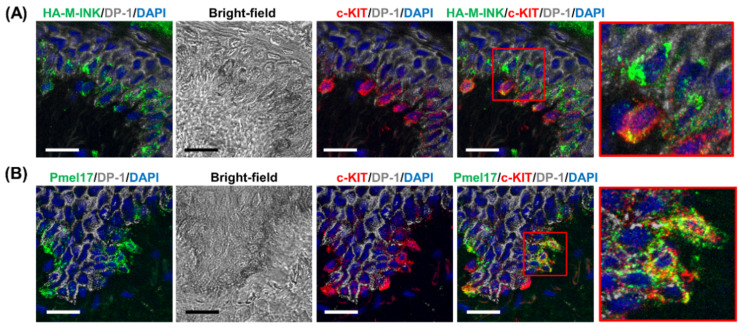
Immunostaining images of HA-M-INK and Pmel in human skin sections. (**A**) Skin section obtained from a phototype IV subject was stained for HA-M-INK (green), c-KIT (melanocyte marker, red), DP-1 (keratinocyte marker, gray), and DAPI (blue). (**B**) The sample was stained for Pmel (premelanosome marker, green), c-KIT, DP-1, and DAPI. The insets are magnified views of the boxed areas. HA-M-INK-positive melanin was observed not only in melanocytes but also in keratinocytes. By contrast, Pmel staining was mainly observed in melanocytes, and only a few signals were present in keratinocytes. Scale bars = 20 µm.

**Figure 5 ijms-21-08514-f005:**
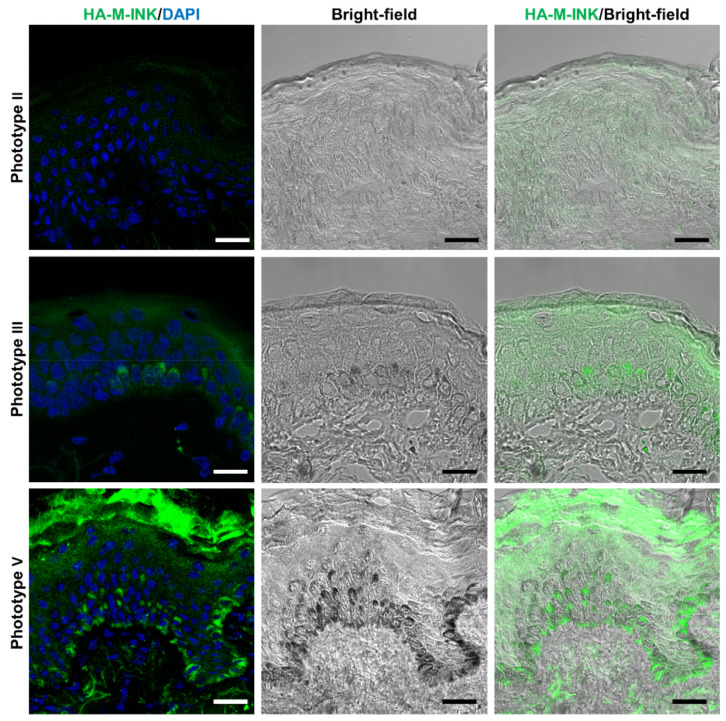
Immunostaining images of HA-M-INK and melanin content in the skin from different skin phototypes. Skin sections of phototypes II (top), III (middle), and V (bottom) were stained for HA-M-INK (green) and DAPI (blue). Phototype II: The sample was mostly negative for HA-M-INK. Phototype III: Melanin was observed as a melanin cap in the spinous layer and in the basal layer, and was colocalized with HA-M-INK. Phototype V: Melanin was observed as a melanin cap, which was widely distributed across the basal and spinous layers, and HA-M-INK-positive signals were well colocalized with the melanin. Scale bars = 20 µm.

**Figure 6 ijms-21-08514-f006:**
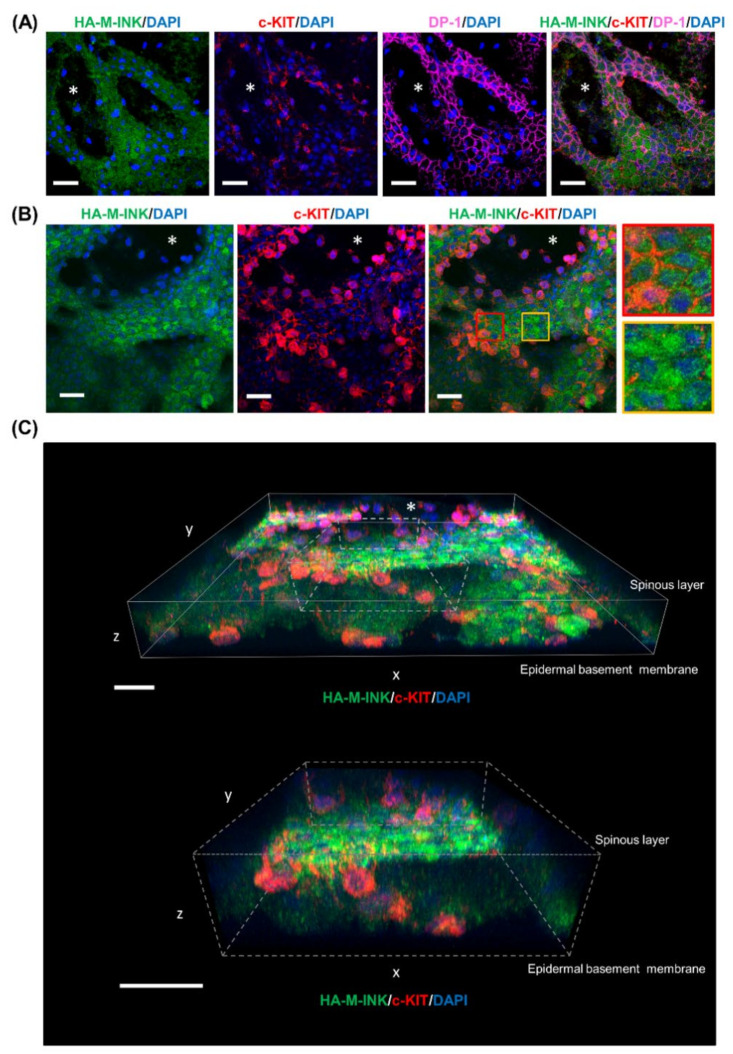
Immunostaining images and three-Dimensional (3D) fluorescence image of epidermal sheets. Epidermal sheets were prepared from a skin sample (phototype V) and stained for HA-M-INK and other markers. Confocal microscopy images were taken from the dermal side and obtained from the epidermal basement membrane to the spinous layer. The asterisks indicate the epidermal structures protruding into the papillary dermis. (**A**) The samples were stained for HA-M-INK (green), c-KIT (red), DP-1 (magenta), and DAPI (blue) and observed in basal layer keratinocytes. (**B**) The samples were stained for HA-M-INK (green), c-KIT (red), and DAPI (blue) and observed from the basal layer side to the spinous layer at 0.5-µm intervals; the images were then superimposed. The insets are magnified views of the boxed areas to show the distribution of melanin in keratinocytes in contact with neighboring (red inset) or non-neighboring (yellow inset) melanocytes. (**C**) The image was constructed three-dimensionally from the images in (**B**). The bottom image is a magnified view of the dotted line areas. Scale bars = 20 µm.

**Table 1 ijms-21-08514-t001:** Base characteristics of skin samples, their skin surface L* values, and fluorescence intensity of HA-M-INK.

No.	Phototype	Age	Sex	Source	L* Value	Fluorescence Intensity of HA-M-INK
1	II	54	Female	Abdomen	77.86	3.69
2	III	50	Female	Breast	73.78	9.41
3	V	27	Female	Abdomen	43.37	12.68

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
