# Peer review of "A Novel Method for Visualizing Melanosome and Melanin Distribution in Human Skin Tissues"

_ijms, 2020, doi:10.3390/ijms21228514_

Round 1
Reviewer 1 Report
good paper, well designed to offer valuable model for the melanin detection tools involving molecular recognition -
notably binding seem to be tightly linked to melanin pigment and this reviewer to consider using chemically pure eumelanin sample for further study development instead of commercial samples
Author Response
good paper, well designed to offer valuable model for the melanin detection tools involving molecular recognition –
notably binding seem to be tightly linked to melanin pigment and this reviewer to consider using chemically pure eumelanin sample for further study development instead of commercial samples.
We thank the reviewer’s very positive comments and helpful suggestion. We also think that it is be interesting to test the interaction between M-INK and chemically pure eumelanin. In the current manuscript, however, we believe that such data is unnecessary. We just wanted to show that M-INK can bind to purified melanin in vitro. We would like to test the interaction of M-INK with chemically pure eumelanin and pheomelanin in our future study.

Reviewer 2 Report
To the authors:
In general, the manuscript describes a novel method to detect melanin and melanosomes in epidermis. The authors are required to answer the comments below;
Comments 1: HA-M-INK staining of melanin cap of keratinocytes at basal and suprabasal layer is not clear. More clear photographs are required.
Comments 2: Melanosomes in keratinocytes are degraded gradually according to their differentiation process moving from basal layer to outer surface of epidermis. How clearly new probe can detect melanin degraded?
Comments 3: Melanosome size differs among skin types, larger in black skin and smaller in white, photo type 1 skin. Authors should show the difference of the melanosome size by HA-M-INK probe staining.
Comments 4: Melanosomes of photo type 1 and 2 are reported to make cluster, whereas, melanosome of type 5 and 6 do not make cluster. How clearly this new probe can show melanosome life dynamics ?
Comments 5: Horny layer seems to be positively stained by HA-M-INK staining, and that of photo type 5 is very strong, suggesting that melanin in horny layer is recognized by this staining.
Why the authors hesitate to mention that melanin remains at horny layer in skin type 5, but not in skin type 2, and type 3?
Comments 6: The effect of PAR-2 activity on melanosome transfer from melanocytes to keratinocytes should be studied by this new probe.
Comment 7: Composition of melanin, eumelanin and pheomelanin may give some effect on the staining ability of HA-M-INK probe staining. The effect should be described in the text.
Comment 8: The number of skin used to determine the difference of staining characteristics of melanosomes among skin types is too small to come to conclusion of skin type effect on melanosomes contents.
Author Response
[Response to the reviewer #2]
In general, the manuscript describes a novel method to detect melanin and melanosomes in epidermis. The authors are required to answer the comments below;
We appreciate the reviewer’s critical assessment of our manuscript. We have addressed the reviewer’s concerns as follows.
- HA-M-INK staining of melanin cap of keratinocytes at basal and suprabasal layer is not clear. More clear photographs are required.
Low resolution of the images may be due to the conversion to pdf. In the revised manuscript, we have also uploaded the original high resolution images. Please see these original images.
- Melanosomes in keratinocytes are degraded gradually according to their differentiation process moving from basal layer to outer surface of epidermis. How clearly new probe can detect melanin degraded?
We also think that this is an interesting and important question. However, our M-INK probe can detect the amount of melanin, but cannot monitor the degradation status of melanin. Thus, we did not discuss this point in the revised manuscript.
- Melanosome size differs among skin types, larger in black skin and smaller in white, photo type 1 skin. Authors should show the difference of the melanosome size by HA-M-INK probe staining.
Although we can detect individual melanosomes in cultured B16 cells, we were unable to observe a single melanosome in the immunohistochemical samples due to the relatively low resolution of the thick tissue samples. We sincerely hope that the reviewer would understand such technical difficulty.
- Melanosomes of photo type 1 and 2 are reported to make cluster, whereas, melanosome of type 5 and 6 do not make cluster. How clearly this new probe can show melanosome life dynamics?
As described in the response to the comment #3, it is extremely difficult to observe individual melanosomes in tissue samples by conventional confocal microscope. We would like to test whether M-INK can also apply to immuno-EM in our future study to analyze melanosome clusters in tissue samples.
- Horny layer seems to be positively stained by HA-M-INK staining, and that of photo type 5 is very strong, suggesting that melanin in horny layer is recognized by this staining.
Why the authors hesitate to mention that melanin remains at horny layer in skin type 5, but not in skin type 2, and type 3?
We also think that M-INK signals in horny layer would be melanin. According to our experience, however, some antibodies non-specifically recognize horny layer in tissue samples. So, that’s why we used the weak expression. In the revised manuscript, we have added this concern (lines 285-286, page 9).
- The effect of PAR-2 activity on melanosome transfer from melanocytes to keratinocytes should be studied by this new probe.
This is an interesting experiment, but it is clearly outside the scope of this study. We will try it in our future study.
- Composition of melanin, eumelanin and pheomelanin may give some effect on the staining ability of HA-M-INK probe staining. The effect should be described in the text.
M-INK is likely to recognize black insoluble eumelanin (Fig. 2), but we have not yet tested whether M-INK binds to pheomelanin. Therefore, we do not want to describe staining of pheomelanin in skin tissues in the current manuscript.
- The number of skin used to determine the difference of staining characteristics of melanosomes among skin types is too small to come to conclusion of skin type effect on melanosomes contents.
We agreed with the reviewer. We wanted to test more samples, but due to the COVID-19 pandemic, we analyzed a limited number of samples. Thus, we toned down the expression and added the following sentence in the revised manuscript: “we need to test more samples to support this conclusion” (line 193, page 7 and lines 279-280, page 9).
